# Wildfires increasingly impact western US fluvial networks

Grady Ball [1,6], Peter Regier [2,5,6], Ricardo González-Pinzón [2✉], Justin Reale [3] & David Van Horn [4✉]

Wildfires are increasing globally in frequency, severity, and extent, but their impact on fluvial networks, and the resources they provide, remains unclear. We combine remote sensing of burn perimeter and severity, in-situ water quality monitoring, and longitudinal modeling to create the first large-scale, long-term estimates of stream+river length impacted by wildfire for the western US. We find that wildfires directly impact ~6% of the total stream+river length between 1984 and 2014, increasing at a rate of 342 km/year. When longitudinal propagation of water quality impacts is included, we estimate that wildfires affect ~11% of the total stream+river length. Our results indicate that wildfire activity is one of the largest drivers of aquatic impairment, though it is not routinely reported by regulatory agencies, as wildfire impacts on fluvial networks remain unconstrained. We identify key actions to address this knowledge gap and better understand the growing threat to fluvial networks, water security, and public health risks.

[1] Water Resources Graduate Program, University of New Mexico, Albuquerque, NM, USA. [2] Department of Civil, Construction & Environmental Engineering, University of New Mexico, Albuquerque, NM, USA. [3] U.S. Army Corps of Engineers, Albuquerque District, Albuquerque, NM, USA. [4] Department of Biology, University of New Mexico, Albuquerque, NM, USA. [5] Present address: Pacific Northwest National Laboratory, Richland, WA, USA. [6] These authors contributed equally: Grady Ball, Peter Regier. ✉email: gonzaric@unm.edu; vanhorn@unm.edu

Wildfires are increasing in frequency, severity, and extent across the globe[1,2]. The 2019–20 wildfire season in Australia was the worst in recent history[3], wildfire activity in the Amazon is threatening to shift the region from a sink to a source of carbon, potentially releasing as much as 17 Pg of $CO_2$ into the atmosphere by 2050[4], and in 2019, wildfires burned across the Arctic at an unprecedented scale[5]. Current wildfire models predict that the prevalence of wildfire and associated damage will continue to increase due to anthropogenic climate change and forest management practices[6–10].

In the western United States (US), growing human populations are encroaching on previously undeveloped areas and expanding the fire-prone wildland-urban interface[11,12], resulting in unprecedented vulnerability and damage. For example, the 2018 California wildfire season, the previous worst on record, burned ~8000 km$^2$, claimed 100 lives, damaged >24,000 structures, and prompted >$2 billion in insurance claims[13,14]. It was recently superseded by the 2020 wildfire season[15], which burned more than 17,000 km$^2$. Since forested watersheds supply drinking water for approximately two-thirds of the western US[16] and large portions of the rest of the world with an estimated value of $4.3 trillion[17], wildfire damage to aquatic systems in forested watersheds represents a significant and costly threat to water security, both regionally for the western US and globally[18–20].

There is growing evidence that wildfires trigger cascading impacts in fluvial networks over a range of spatiotemporal scales[18,21–24]. Wildfires originate on hillslopes and cause decreased infiltration capacity and groundwater recharge[25–27], increased overland flow[22,28], reduced flood attenuation capacity by riparian vegetation[29], increased snow ablation[30], and higher frequency of landslides, avalanches, and debris flows[31,32]. Post-fire precipitation events mobilize wildfire-generated material from terrestrial ecosystems into streams and rivers within burned areas, which in turn drain into larger fluvial networks. Along these pathways, surface water quality drastically changes due to increased fluxes of ash, sediments, nutrients, carbon, and metals, commonly causing exceedances to limits set by the World Health Organization's safe drinking water standards[17,33–36], and increasing costs associated with irrigation and drinking water supply[35,36]. Wildfire disturbances also contribute to at least ten of the top twenty most critical disturbances listed in the US Environmental Protection Agency's Clean Water Assessment (US EPA CWA[37]), i.e.,: elevated sediments, nutrient enrichment, organic enrichment and oxygen depletion, elevated temperature, elevated metal concentrations, habitat alterations, elevated turbidity, flow alterations, elevated salinity and/or total dissolved solids, and changes to pH and conductivity[17,38,39].

Despite the growing threat of wildfire impacts to water quality and ecosystem services that protect public health, we lack quantitative estimates of the stream + river length directly impacted by wildfires (i.e., within burned areas) and how far downstream wildfire-driven water quality impacts propagate. Addressing these knowledge gaps is important to (1) alert downstream communities about the risk of exposure to wildfire-related contaminants, (2) anticipate the range of potential impacts of wildfire to downstream water quality, (3) develop risk maps to mitigate the damage of property and infrastructure in fire-prone areas, (4) apply water treatment techniques capable of removing wildfire-specific physical and chemical pollutants, (5) implement post-fire emergency watershed rehabilitation techniques to reduce the movement of sediment, burned vegetation, nutrients, metals, and other contaminants from hillslopes to streams, and (6) design effective long-term post-fire restoration projects to increase revegetation, water filtration, and sediment retention at watershed scales.

In this study, we estimate the Stream + river Length impacted within Burned Areas ($SL_{BA}$) across the western conterminous US

from 1984 to 2014. We explore temporal $SL_{BA}$ trends and relationships to the burned area as a function of established ecoregions[40], since many drivers of wildfire, such as climate regime[41], drought[42], and snowpack[43], are ecoregion-specific. Next, we use high-frequency in-situ sensor data capturing the spatial propagation of post-fire water quality disturbances in the Rio Grande (New Mexico, US) to estimate the Stream + river Length: Longitudinal Extent ($SL_{LE}$) impacted by an individual wildfire as a case study, and upscale this approximation to the study region. Finally, we identify three essential action items for scientists, land and water managers, and funding agencies to better assess the propagation of wildfire-related water quality impacts across forested watersheds.

## Results and discussion

**Stream + river length impacted within burned areas ($SL_{BA}$).** We performed a geospatial analysis combining wildfire extent, burn severity, hydrography, and ecoregion layers to calculate the first estimate of $SL_{BA}$ in different ecoregions in the western US (see Methods). Wildfires included in the dataset ($n = 7677$, Supplementary Table 1) span 9 ecoregions in 11 states (Fig. 1). The total $SL_{BA}$ (324,080 km, Supplementary Table 1) represents 5.7% of total stream + river length for the study area, varying between ecoregions from 0.1% (Marine West Coast Forest) to 12.4% (Mediterranean California). Kernel density plots demonstrate considerable latitudinal and longitudinal variability in $SL_{BA}$, with maxima between 40°N and 45°N, and 110°W and 120°W (Fig. 1). In this area, composed primarily of Cold Desert and Western Cordillera ecoregions, streams are disproportionately impacted by wildfire. Ecoregion-specific density distributions highlight differences in the impact of a single ecoregion on the cumulative spatial density of streams impacted by wildfire. For instance, Marine West Coast Forests, the ecoregion with the lowest wildfire activity (Supplementary Table 1), has clearly defined latitudinal and longitudinal peaks, but does not noticeably alter the general density curve (Fig. 1). In contrast, the similarly sized but more wildfire-prone Upper Gila Mountains ecoregion drives a clear second latitudinal peak near 33°N. Thus, ecoregion-specific parameters drive differences in how wildfire impacts streams.

We found a highly significant increasing trend ($p < 0.0001$) for $SL_{BA}$ over the study period, with an average increase of 342 km/ year (Fig. 2A). Because not all wildfires are mapped in Monitoring Trends in Burn Severity (MTBS)[41], and our analysis excludes some wildfires (see Methods), these trends represent underestimates. Within individual ecoregions, annual $SL_{BA}$ varied by orders of magnitude (Fig. 2B), from 0 km (color-coded as gray, and observed for multiple ecoregions and years) to 14,046 km in 2012 in the Cold Deserts ecoregion (Supplementary Table 2). On average, for all ecoregions combined, $SL_{BA}$ equaled 10,454 km/ year, with an annual maximum of 34,112 km impacted in 2012 (Supplementary Table 2). While the Mediterranean California ecoregion contains only 6% of total stream + river length (Supplementary Table 1), it accounted for ~14% of $SL_{BA}$ across the study period, and 54% of all $SL_{BA}$ in 1997 (Supplementary Table 2). Three other ecoregions (Cold Deserts, West-Central Semiarid Prairies, and Western Cordillera) accounted for more than half of the total annual $SL_{BA}$ (maxima of 69% in 1985, 52% in 1991, and 68% in 1987, respectively). In contrast, wildfires occurred within the Marine West Coast Forest ecoregion in only 13 of the 31 years of the study (Fig. 2), with a maximum annual $SL_{BA}$ of 93 km in 2008 (<1% of annual $SL_{BA}$). As Fig. 2 demonstrates, the extreme inter-annual and inter-ecoregion variability in $SL_{BA}$ represents a significant obstacle for those in charge of allocating resources to monitor and mitigate impacts of wildfires on fluvial networks.

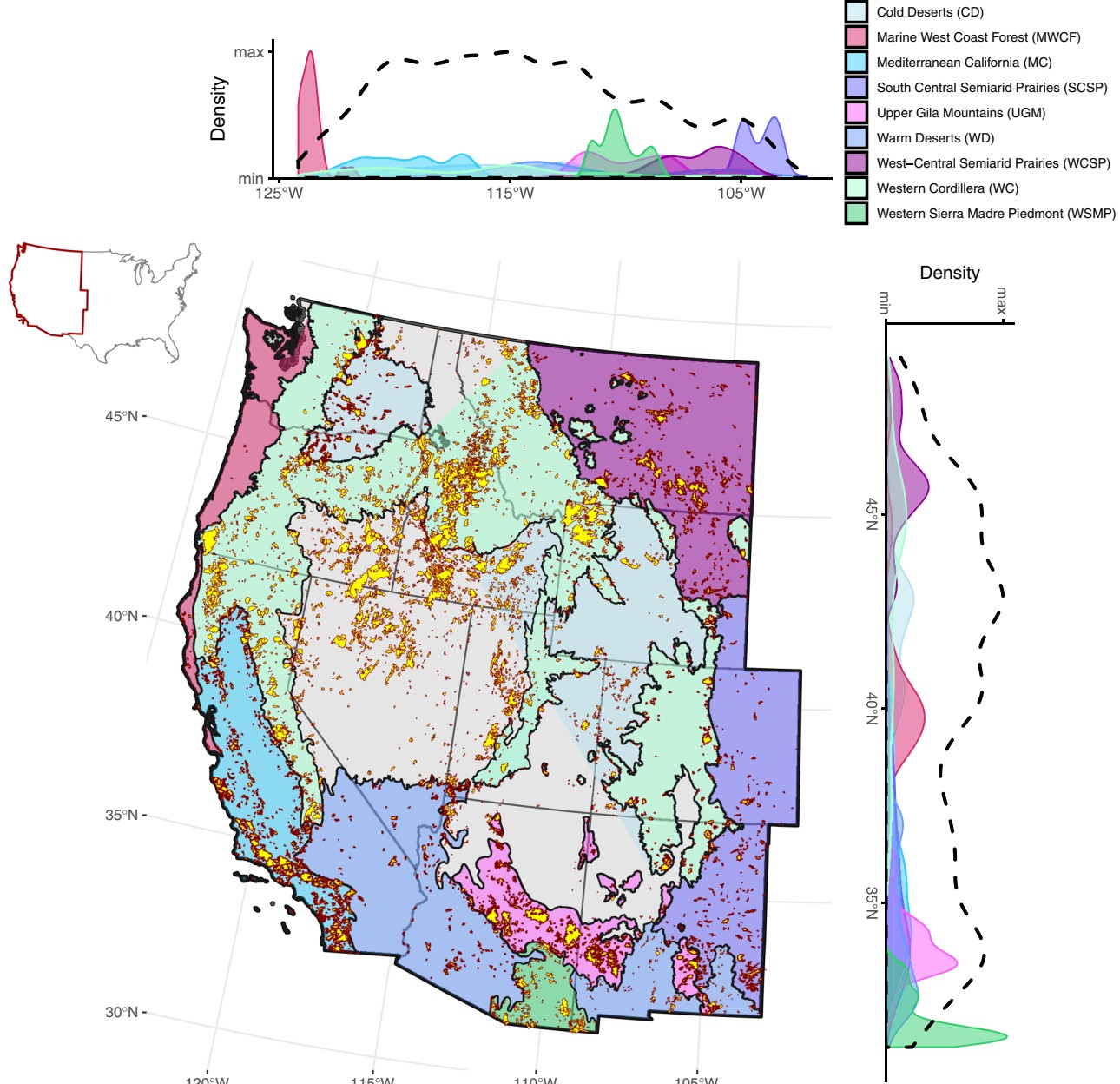

**Fig. 1 Spatial distribution of wildfires across the western conterminous US.** Data span from 1984–2014. Burn scars are filled in yellow and outlined in red. Kernel density plots show the latitudinal and longitudinal distribution of Stream+river Length impacted within Burned Areas ($SL_{BA}$) for the full dataset (dashed lines) and by ecoregion (shaded regions).

We next explored the relationship between area burned and $SL_{BA}$ by year (data in Fig. 2) and ecoregion (Supplementary Table 3). For all ecoregions combined, a strong, highly significant correlation ($R^2_{adj} = 0.91$, $p < 0.0001$) with a slope of 0.98 km/km² suggests a conservative approximation that 1 km² burned yields ~1 km of $SL_{BA}$. All 9 ecoregion-specific best-fit lines for $SL_{BA}$ versus area burned were significant (8 were highly significant ($p < 0.0001$), $R^2_{adj} \geq 0.82$, with the relationship for the Marine West Coast Forest as the only exception: $R^2_{adj} = 0.54$, $p = 0.0025$, Supplementary Table 3). Differences in slopes between ecoregions spanned a factor of ~3, meaning a wildfire footprint of 1000 km² is predicted to directly affect only 470 km of stream + river length if occurring in the Warm Deserts ecoregion, but 1590 km in the Western Sierra Madre Piedmont ecoregion. When the Marine West Coast Forest ecoregion was excluded, a strong, moderately

significant correlation was present between the drainage network density (total stream + river length:total area) and the $SL_{BA}$: burned area ratio ($R^2_{adj} = 0.66$, $p = 0.0091$, $n = 8$, Supplementary Fig. 1). This suggests that the density of streams burned generally correlated with the density of streams in an ecoregion. However, ecoregion-specific factors (e.g., the Marine West Coast Forest ecoregion receives 210% of the annual precipitation of the next wettest ecoregion included in the study) created an outlier. Thus, the relationship between area burned and $SL_{BA}$ is likely a function of regional factors, including geomorphology, vegetation, and climate, suggesting that numerous factors control the extent to which wildfires impact aquatic resources in a given location.

**Downstream propagation of wildfire-driven water quality disturbances.** Our $SL_{BA}$ calculations represent a conservative

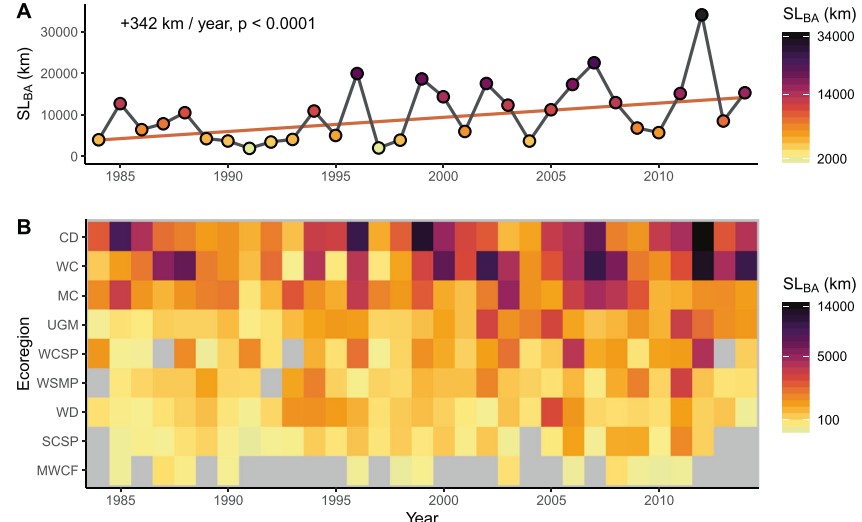

**Fig. 2 Stream+river length impacted within burned areas (SL$_{BA}$). A** The trend of SL$_{BA}$ for all ecoregions combined through the study period. **B** Trends divided into individual ecoregions, ordered from smallest number of wildfires (bottom) to largest (top), show large variations in annual SL$_{BA}$ patterns. Gray squares indicate year/ecoregion combinations where no wildfires were reported that match the Monitoring Trends in Burn Severity criteria. The ecoregion acronyms are Cold Deserts (CD), Marine West Coast Forest (MWCF), Mediterranean California (MC), South Central Semiarid Prairies (SCSP), Upper Gila Mountains (UGM), Warm Deserts (WD), West-Central Semiarid Prairies (WCSP), Western Cordillera (WC), Western Sierra Madre Piedmont (WSMP).

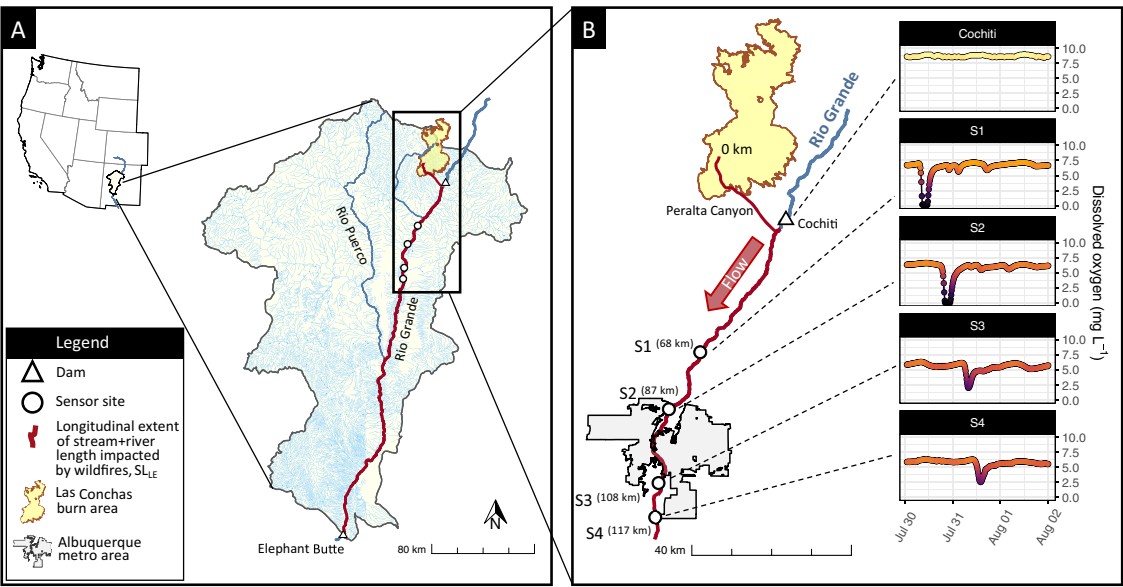

**Fig. 3 Longitudinal propagation of disturbances after the Las Conchas wildfire. A** Locations of four water quality sondes deployed along the Rio Grande (New Mexico, US), which measured water quality dynamics pre- and post-fire. **B** Example of the measured downstream progression of a dissolved oxygen (DO) sag originating from the headwaters of Peralta Canyon; this and three other DO sag events were used to predict the longitudinal extent of the stream +river length impacted by wildfire disturbances (SL$_{LE}$) beyond the S4 sensor (see panel **A**). A DO sensor located 220 m below Cochiti Dam and 4 km upstream of the Peralta Canyon confluence exhibited no DO sag, likely because the reservoir has long residence times and a turbulent release that favors fast reaeration.

estimate of stream + river length affected by wildfire as they ignore potential downstream propagation of aquatic disturbances driven by post-fire precipitation events. As a first attempt to address this limitation, we used water quality data collected in the Rio Grande to estimate the longitudinal extent of water quality impacts after the 2011 Las Conchas wildfire, the second largest wildfire recorded in New Mexico, US, which burned 630 km$^2$ with 388 km of SL$_{BA}$. The dataset, originally presented in Dahm et al.[22] and used here, represents one of the most spatially and temporally explicit records of pre- and post-fire water quality available to date, as it registered the variation of water quality parameters along ~50 km of the Rio Grande, and over multiple years before and after the wildfire. The Las Conchas wildfire burned in the Western Cordillera ecoregion and propagated through portions of the Rio Grande in both the Western Cordillera and the Cold Desert ecoregions. Post-fire monsoon precipitation events mobilized ash and other burned materials downstream through intermittent channels, entering the Rio

Grande downstream of Cochiti Dam, primarily via the Peralta Canyon watershed (Fig. 3A).

Four multi-parameter water quality sondes that were located 68–117 km downstream of the headwaters of Peralta Canyon fortuitously captured multiple severe water quality disturbances following monsoon precipitation events, as evidenced by dissolved oxygen (DO) sags (Fig. 3B). The recorded data showed that between July and August 2011, 11 post-fire runoff pulses (~10–20 m$^3$s$^{-1}$) from monsoon precipitation events lowered DO at S1 to levels below the water quality standard established for local aquatic life (5.4 mg/L), with four events reaching short-lived anoxia (~1–4 h)[22]. These types of episodic DO sags below the water quality standard persisted through 2014 at S1[23,44]. Outside of the monsoon season, DO concentrations at S1 remained >6 mg/L prior to and following the wildfire. In contrast, DO levels directly below Cochiti Dam remained above 7.0 mg/L (Fig. 2B) through July and August 2011[22], suggesting that most of the DO sags were triggered by runoff draining Peralta Canyon. Besides causing DO sags, these excursions transported large quantities of ash and sediment, forcing a 2-month shutdown of the City of Albuquerque's (population of ~560,000) only surface water intake, which provides ~70% of the drinking water supply[45].

We analyzed the longitudinal propagation of distinguishable DO sags generated after four monsoon runoff events that occurred after the Las Conchas wildfire and fitted an exponential model to estimate the extent of these wildfire impacts on water quality. We chose this model because it provided a strong statistical prediction of our dataset, and a mechanistic interpretation of advection and reaction processes in rivers[46]. Our model estimated that DO sags >0.5 mg/L (used as our impact threshold) persisted 344 km downstream of the Peralta Canyon headwaters ($R^2 = 0.74$, $n = 13$, Supplementary Fig. 2), which is equivalent to 89% of the SL$_{BA}$ for the Las Conchas wildfire (388 km). Since DO is reincorporated via reaeration during transport, we predict that the longitudinal propagation of more conservative signals (i.e., non-limiting nutrients, metals, or ash) extend farther than DO sags. As such, we suggest a conservative estimate of the longitudinal extent of stream + river length impacted by wildfire disturbances (SL$_{LE}$) as SL$_{LE}$~SL$_{BA}$. As an independent verification of this estimation, we repurposed Horton's law of stream length[47] to create a mathematical framework commensurate with our data availability to validate our estimates of SL$_{LE}$. This work resulted in a new longitudinal model to estimate SL$_{LE}$ downstream of burned areas. When applied to our dataset, it estimated that the disturbances could propagate to an 8th order stream (i.e., the Rio Grande downstream of its confluence with the Rio Puerco, 197 km downstream of the headwaters of Peralta Canyon, Fig. 3A; see Methods). Paradoxically, even with a spatial network of high-resolution water quality sondes deployed downstream of the Las Conchas wildfire during post-fire runoff events, we still lack information to accurately constrain the full extent of wildfire-driven water quality impacts to downstream aquatic systems in the Rio Grande watershed beyond this first approximation.

Using the SL$_{LE}$ estimates from the analysis of the Las Conchas wildfire, we estimated the total stream + river length affected in each ecoregion of the western US over the study period by combining our estimates of SL$_{LE}$ and SL$_{BA}$. Wildfires impacted an average of ~11% of all stream + river length within the study area between 1984 and 2014 (Fig. 4). In Mediterranean California and Western Sierra Madre Piedmont ecoregions, our approach indicates ~25% and ~21% of total stream + river length was impacted by wildfire during the study period, respectively. In contrast, <1% of stream + river length was impacted in the Marine West Coast Forest. Based on the US EPA CWA for 2012, the year with the highest wildfire stream impacts in our dataset

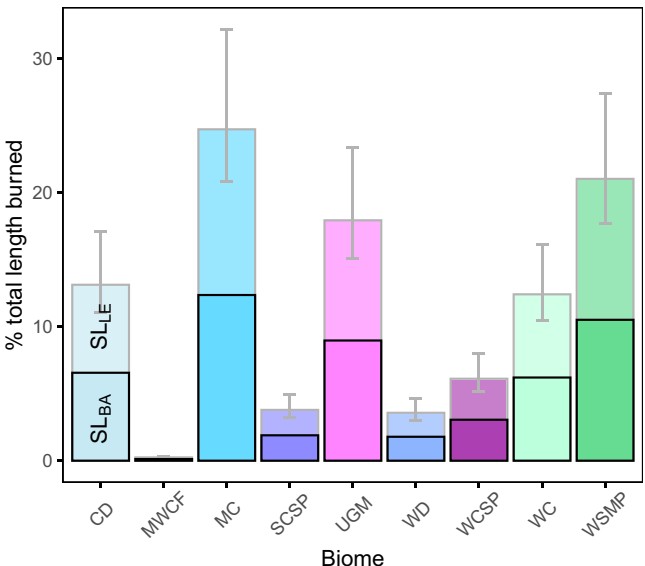

**Fig. 4 Percentage of total stream+river length burned by ecoregion.** Darker bars outlined in black on the bottom indicate the percent calculated from the quantification of the stream+river length impacted within burned areas (SL$_{BA}$), and lighter bars on the top represent our estimation of the longitudinal extent of stream+river length impacted by wildfire disturbances (SL$_{LE}$). Error bars represent the 95% confidence interval associated with the fit-line presented in supplementary Fig. 2. The ecoregion acronyms are Cold Deserts (CD), Marine West Coast Forest (MWCF), Mediterranean California (MC), South Central Semiarid Prairies (SCSP), Upper Gila Mountains (UGM), Warm Deserts (WD), West-Central Semiarid Prairies (WCSP), Western Cordillera (WC), Western Sierra Madre Piedmont (WSMP).

(SL$_{BA}$ + SL$_{LE}$ = 68,224 km, see Supplementary Table 2), wildfire would represent the 8th largest single source of water quality impairment to streams and rivers. As the US EPA CWA covers all US states and territories, and our analysis is limited to the conterminous western US, wildfires are likely ranked even higher as a primary source of water quality impairment. Our findings highlight both the importance of explicitly including wildfires in regional and national water quality assessments, and the urgent need to collect information to constrain their impact on water quality. We recognize that this first approximation does not account for variability in watershed and stream conditions that could influence the longitudinal propagation of wildfire impacts, or for inter-ecoregion variability (i.e., Supplementary Fig. 1). Nevertheless, our estimates are a preliminary assessment that can be temporarily used to address problems requiring an answer to the question: how far do wildfire disturbances propagate along fluvial networks? More importantly, we emphasize that the limitation of our estimates serves as a wake-up call for the scientific community to start gathering longitudinal data to improve our predictive ability to assess disturbance propagation.

**Incorporating streams and rivers into fire science.** Wildfires are inherently complex, highly disruptive events that increasingly threaten lives, property, and natural resources throughout the western US and much of the world. These disturbances mobilize sediments, nutrients, carbon, metals, and other environmentally relevant materials, affecting water quality dynamics not only along fluvial networks but also in downstream lacustrine, estuarine, and marine environments. Our calculations suggest that ~11% of all stream + river length across the western US was impacted by wildfire in the last three decades, with as much as

25% impacted in individual ecoregions. Although the estimated impacts of wildfires on streams in Fig. 4 vary considerably across ecoregions, likely explained by differing ecoregion and watershed characteristics (e.g., drainage network density and connectivity, aridity, geomorphology, and precipitation levels) and wildfire features (e.g., severity and spatial extent), these findings represent, to our knowledge, the first attempt to quantify wildfire impacts on streams and rivers at a regional or continental scale. We attribute the lack of previous estimates to the notable absence of water quality data collected from multiple sensors located over spatial scales commensurate with the propagation of wildfire disturbances (i.e., over hundreds of kilometers). Thus, additional studies of the impacts of wildfires on water quality dynamics and longitudinal propagation are urgently needed to more thoroughly assess the spatial and temporal extent to which fluvial networks are impacted, and to develop stronger predictive tools to assess wildfire impacts on water quality to improve mitigation and resilience strategies. We foresee that the next generation of studies capable of transforming our understanding of the spatiotemporal response of watersheds to natural and human-caused disturbances require a new funding mechanism centered around preparation and readiness. Here, we identify three action items aimed at transforming the acquisition of water quality data to support a dynamic approach to incorporating aquatic systems into wildfire science:

1. Invest funding in preparation and readiness: Wildfires are a top-10 contributor to water quality impairment and represent a significant threat to water security[37,48–50]. As such, wildfire impacts to streams and rivers need to be prioritized for local, state, federal and international funding, and included as a component of wildfire preparedness plans to reduce vulnerability and promote resilience[51]. Because wildfires behave unpredictably and evolve rapidly[52], the current avenues funding wildfire research, which require weeks to months to obtain resources, are inadequate. We advocate for the creation of funding mechanisms to equip rapid-response teams proximal to wildfire-prone areas to respond immediately to wildfires. Teams should be continuously funded so they are ready to capture first-flush watershed responses whenever and wherever wildfires occur.

2. Increase focus on capturing longitudinal behavior: The downstream propagation of wildfire impacts is severely understudied. Even the state-of-the-art sensor network fortuitously available along the Rio Grande did not capture the full extent of the propagation of the aquatic disturbances generated after the Las Conchas wildfire (Figs. 3 and S2). Thus, we must begin to incorporate 'dynamic' monitoring approaches focused on longitudinal data collection that supplement traditional 'stationary' ecological monitoring strategies (e.g., CZO and LTER) to fully characterize the extent of wildfire-driven water quality excursions along fluvial networks. Dynamic monitoring should be the priority of rapid-response teams.

3. Incorporate high-frequency data in environmental monitoring: Wildfire related water quality disturbances occur rapidly and over a short duration, thus, high-resolution and real-time data are essential for capturing temporally dynamic and ecologically significant events. Therefore, it is crucially important to employ recent technological advances in aquatic monitoring tools to document the impacts of wildfires. These include multi-parameter sensors[23,53], unattended sampling methods[54,55], autonomous vehicles[56,57], near-real-time modeling[58], and machine learning[59,60], to collect and interpret high-frequency data in near real-time.

## Methods

**Datasets.** We used the National MTBS Burned Area Boundaries dataset and Burn Severity Mosaics (1984–2014, https://www.mtbs.gov/direct-download) to calculate the area burned and burn severity for each wildfire. We used the 1:24,000 National Hydrography Dataset (NHD, https://nhd.usgs.gov/data.html) bounded by MTBS burned areas to calculate stream+river length impacted with burned areas ($SL_{BA}$) for each wildfire, and EPA Level 2 Ecoregions boundaries (https://www.epa.gov/eco-research/ecoregions-north-america) to define ecoregions[40]. We included all catchments and stream orders in the NHD layer within the study region in our analysis. The number of wildfires, area burned, and $SL_{BA}$ calculated from these datasets are summarized by ecoregion in Supplementary Table 1.

**Geospatial analysis.** Geospatial data preparation was conducted in ARCMap 10.4 (Environmental Systems Research Institute, Redlands, CA), and spatial data analysis was conducted in Python using the Pandas data analysis library (version 0.25.1). Briefly, we utilized ARCMap's Model Builder to create a geospatial workflow to iteratively process multiple datasets. MTBS Burn Severity Mosaics were converted from rasters to polygons, and limited to low, medium, and high burn severity to remove areas coded as unburned, increased greenness, or masked. After filtering for severity, any wildfires burning <~4.1 km² were removed from further analysis. Next, $SL_{BA}$ values were derived from NHD data and vectorized MTBS burned area data. For wildfires spanning multiple ecoregions, we assigned ecoregion by largest area burned, so a single ecoregion was assigned to each unique wildfire ID. This yielded 0.1–3.8% of burned area in a given ecoregion being attributed to a different one. Since the dataset was downloaded in 2017, and the MTBS dataset is frequently updated, our dataset represents conservative estimates of area burned and $SL_{BA}$.

**Statistics.** Statistical analysis and plotting were primarily conducted in R v3.5.1 (R Core Team). We used the non-parametric Theil-Sen method[61] to estimate regression slopes for time-series, which is more robust to outliers than common parametric methods. Linear regression models were established for area burned versus $SL_{BA}$ in R by binning the sum of each parameter for each year in each biome. A two-component exponential decay model was fitted for change in DO (calculated by subtracting the minimum value from the pre-sag baseline value for each event at each site) in JMP v13.2.0 (SAS Institute, Cary, NC) using the Fit Curve platform. Four distinct water quality disturbance events were identified by DO sags occurring at all sites with usable data. All events exhibited consistent temporal lags between sites and consistent sag shapes across sites. Only storms with clear sags for at least three sites were included. For statistical clarity, we define weakly significant as $p < 0.05$, moderately significant as $p < 0.01$, and highly significant as $p < 0.0001$.

**Validating longitudinal stream + river length ($SL_{LE}$) assumptions.** The longitudinal propagation of wildfires through fluvial networks can be modeled through a combination of solute and sediment transport models. However, such models require information about boundary conditions and spatiotemporal variability of parameters and geomorphology that currently available data cannot constrain[62]. Instead, we repurposed Horton's law of stream length[47] to create a mathematical framework commensurate with current data availability to validate our estimates of $SL_{LE}$. Horton proposed that the length ratio $R_L = \bar{L}(\omega + 1)/\bar{L}(\omega)$ followed a predicable scaling pattern in fluvial networks, where $\bar{L}(\omega)$ is the arithmetic average of the length of streams of order $\omega$ and $1.5 < R_L < 3.5$, averaging 2. We used a geometric expansion applicable to in-series connections to project this law and estimate $SL_{LE}$ downstream of a burned area affecting a stream order $\omega_0$:

$$SL_{LE} = \sum_{i=\omega_0}^{\omega} \bar{L}_{\omega_0} \cdot R_L^{(i-1)} = \bar{L}_{\omega_0} \left( \frac{1 - R_L^{\omega}}{1 - R_L} \right) \qquad (1)$$

Knowing that the Las Conchas wildfire burned the headwaters of the Peralta Canyon tributary, including 1st order streams ($\omega_0 = 1$) in a watershed with $\bar{L}(1)\sim1.3$ km[63], and using our estimates of $SL_{LE}\sim344$ km, $\omega = 8$ if $R_L = 2$. This means that a disturbance caused in a 1st order stream could impact up to an 8th order stream in our fluvial network. Thus, our calculations using Horton's law of stream length, which indicates propagation past the confluence of the 7th order Rio Grande with the Rio Puerco (197 km downstream of the headwaters of Peralta Canyon, Fig. 3A), are consistent with our estimates of $SL_{LE}$ (Fig. 3A). We note that Dahm et al.[22] reported that Cochiti Dam (upstream of the confluence with Peralta Canyon) removed wildfire-associated water quality signatures (see also Fig. 3B), even when pulses were observed upstream of the reservoir, indicating that impoundments can strongly alter the propagation of post-fire water quality impacts.

## Data availability

The data generated and analyzed during this study are available in the repository published by Ball et al.[64].

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

## Acknowledgements

Funding was provided by the National Science Foundation (HRD-1720912, HRD-1914490, CBET-1707042, DEB-1748133, and DEB-1440478) and the US Army Corps of Engineers' Upper Rio Grande Water Operations Model (W912HZ-14-2-0014). The authors thank Nick Ward, Allison Myers-Pigg, Stephen Brown, and Bayani Cardenas for providing constructive feedback on early versions of this manuscript.

## Author contributions

G.B. and P.R. contributed to the conceptualization, spatial analysis, generation of figures and tables, writing, review, and editing. J.R. contributed to the contextualization of the study, writing, review, and editing. D.V.H. and R.G.P. conceptualized the study, secured funding, mentored students and post-doc, led the modeling development, and contributed to the writing, review, and editing.

## Competing interests

The authors declare no competing interests.
