## [Peer Review File · Nature Communications]

REVIEWER COMMENTS

Reviewer #1 (Remarks to the Author):

The authors present a study that investigates the impacts of wildfire on fluvial networks in the western US by taking into account stream length impacted by burn events. I have not yet seen any published works that address stream water quality in this way, and thus the proposed work brings a fresh, novel perspective to wildfire science fields. The authors also use a Rio Grande case study to assess downstream propagation effects of wildfire and scale these data to the greater western US region. The authors provide a convincing argument for propagation of wildfire ecological effects in river networks; however, the following comments should be addressed before I can recommend this manuscript for publications in Nature Communications:

Figure 1 – While I appreciate the graphical visualization of regional wildfire distribution in this way, I had a hard time discerning different ecoregions using the current color scheme. Some colors are very similar to one another (e.g., WCSP and WSMP). Please consider changing some of these colors so ecoregions can be easily identified.

L114-120 – The authors refer to Table S3 when describing ecoregion-specific regressions of SLba versus area burned. Some of these relationships were highly significant and I was curious to see how the data were distributed – p values and R2 values only tell part of the statistical story. I recommend that you provide plots for each ecoregion in the supporting information so readers can quickly visualize these relationships.

L146-148 and Figure 3 – Did these DO sags coincide with storm events or were they purely a result of the burn itself? I'm curious to know whether hydrology played a role in driving DO dynamics. If the DO sags coincide with storm events, how does the magnitude of DO depletion compare with non-fire periods?

L152-153 – A more detailed description of the DO exponential decay model is needed. Why did the authors select this particular statistical approach for modeling longitudinal propagation?

L156 – “persisted 344 km downstream” ... What is the error on this number? Please also include error bars or shading in Figure S2. How realistic is the extrapolation of longitudinal distance from the data shown in Figure S2?

L162 – I know it is described in the methods, but Horton's law of stream order needs a brief description and reference here too.

L216 – “Wildfires are a top-10 contributor” ... According to whom? Needs a reference. Could also say a “significant contributor”.

L229-230 – Stationary monitoring are still essential for water quality assessments. Perhaps rephrase to suggest that dynamic monitoring stations are also needed and can leverage data collected continuously from existing fixed-location monitoring stations.

Finally, the impact of predicting longitudinal effects of wildfire is not limited to drinking water, but these findings also have major implications for rivers draining coastal mountainous regions (key geographic feature of California) – The data presented here suggest that downstream propagation of wildfire could alter the amount or composition of carbon, nutrients, or other materials entering estuarine and coastal ocean environments. It would be great if the authors could expand on this in the

conclusion.

Reviewer #2 (Remarks to the Author):

General questions

- What are the noteworthy results? Yes, the manuscript certainly provides noteworthy results on the extent of the western US river networks impacted by wildfire.
- Will the work be of significance to the field and related fields? How does it compare to the established literature? If the work is not original, please provide relevant references. The work is original, and there's not much on this topic. There are hundreds or thousands of manuscripts on the fire impact on communities, sediment dynamics, and biogeochemistry, but there are none assessing the extent. I believe it might impact the field, although a miss a proper discussion of the drivers, consequences and meaning.
- Does the work support the conclusions and claims, or is additional evidence needed? The work support the claims, there's no need for additional evidence.
- Are there any flaws in the data analysis, interpretation and conclusions? - Do these prohibit publication or require revision? No, I believe this is a very good numerical exercise, with no apparent flaws.
- Is the methodology sound? Does the work meet the expected standards in your field? Yes, I believe that the methodology is sound.
- Is there enough detail provided in the methods for the work to be reproduced? Yes, I believe that the methods are properly described and the work is therefore reproducible.

I have a general comment, and some more specific comments. The general is rather conceptual, as I wonder if wildfires are a threat? I mean, they would be only a threat if wildfires were occurring at a frequency, severity and extent higher than what would occur in natural conditions. Fires, as floods, or extreme droughts, are extreme events that have occurred for millennia in western fluvial networks, and the local biodiversity is adapted to it. You need to further discuss this aspect, as perhaps we're recovering a natural disturbance that will allow the communities to recover their pre-settlement composition. In this direction, I would see the fires in the Amazon as a threat, but perhaps not the fires in central Australia or western US. Perhaps wildfires are a short-term threat for water supply, as drinking water treatment facilities relying on surface water might be imperiled, but this is an impact on humans, not on the ecosystem itself.

Specific comments

P2, L40. Nature is an international journal, please use internationally accepted units for surface.

P2, L43. Please, develop more the point on water security. Is it for the changes in the suspended solids (ashes) during the first few months after the fire? Or for what? In terms of water supply, the amount of flowing water might be higher owing to reduced ET in the burned areas. Please, discuss this point. What stated in lines 45-49 is not enough to justify your point.

P4, Figure 2. Is the temporal trend in the SLBA related with increase in the forest cover? With a decrease in rainfall? Regardless of the driver, I suggest you use this figure to discuss the general comment on what is a disturbance and what not? I mean, perhaps we're currently surpassing the natural disturbance fire regime.

Reviewer #3 (Remarks to the Author):

General Comments

The water quality responses to wildfires and the associated changes in ecosystem services from burned watersheds are generally well appreciated. It will be less clear to many readers, why stream length is a relevant metric for gaging them. Stream length is a common reporting structure for water quality impairment but the link to wildfires requires some justification and explanation. Wildfire impacts originate from burning of vegetation and organic soil layers, typically in upland areas and sometimes in riparian vegetation adjacent to streams. Their effects do not originate from changes in streams themselves and evaluating the relative magnitude of effect is typically based on the area burned. Watershed effect of wildfires are usually assessed at the outlet of a burned watershed, so there is need to describe how stream length and catchment areas are related across ecoregions and how those patterns may vary after fire. To help justify how SLBA may have value as a metric of wildfire effects, it would be helpful to include a quick summary of the common sources of stream degradation (nutrients, sediment, temperature, etc.) and how they vary with relevant ecoregional differences. Currently, the authors arrive at this sort of comparison near the end of the paper. This general context should be presented at the very beginning of the paper, so readers will grasp the regulatory relevance for reporting stream length.

The effects of wildfire on downstream water quality relate to the absolute and proportional extent and severity of a given wildfire within a catchment. Wildfires create a complex mosaic of effects since they burn forests and effect watersheds to varying extents depending on the specific patterns and proportional extent of high, moderate, low severity, as well as unburned areas. This is especially relevant to the riparian and stream corridor, since these parts of the landscape may burn at different frequency or severity than upland areas. The authors state that they consider burn severity, but that it was unclear how that information was used. As background, it would be useful to know how the relation between burn severity and fire size varies within ecoregion has that changed with time? What is the relation between proportional burn severity and wildfire extent? Additionally, some characterization of the pre-fire variation in stream density by ecoregions would be a useful precursor before evaluating how wildfire effects stream length regionally and how those changes have varied with time.

The downstream extent of post-fire water quality responses is very interesting. It would be great to know how this relates to the size (proportional extent by fire severity), the size of the watershed, stream discharge, time since fire, etc. The Las Conchas Fire study is interesting, but there is no justification for extrapolating this limited local information across the western US. It would be useful to learn more about whether the observed post-fire DO responses vary with storm characteristics (precip amount and intensity), stream discharge and temperature and also how they differ from pre-fire conditions can DO responses to storm events.

Specific Comments

Line 47 This is incorrect and confusing. Wildfire impacts originate from the burning of vegetation, typically in upland areas surrounding streams and does not originate from changes in streams themselves. This is likely referring to 'the cascading impacts in fluvial networks' caused by wildfire. Nonetheless, wildfires trigger these by altering terrestrial ecosystems and the emergent responses scale with the area of terrestrial ecosystems burned at known levels of wildfire severity. The relations between burn area and stream length may be interesting to consider and report and how se can be proportional area burned After the initial erosion-related responses, post-fire stream nutrient export is the response of lack of plant uptake and accelerated nutrient supply from exposed burned areas. Reporting wildfire effect on a per length basis confuses this issue.

58 A portion of the Rio Grande in New Mexico? Where? How long? Why? The river does not originate or end in New Mexico, so a bit of clarity would be helpful for many readers?

69 Specify the size catchments and minimum stream order included in the survey.

70 Drier areas with few streams and more fires vs wet area with more streams and fewer fires.

73 'hotspot' meaning?

79 Can you report these with regard to stream density and area burned
Fire is would have varying longitudinal extent effects on stream based on this.

90 Information on what fires are/aren't mapped is not clear obvious in Suppl Matter and site selection needs additional clarity.

116-117 This relationship is getting at something useful. Is the difference across regions due to fire, vegetation and fuel relations or merely due to the differences in stream length network density?

126 This would be expected, so how is this insightful?

132 Explain what you mean by resilience and what this statement about ecoregional differences is based upon.

141 Provide a few salient points to support this statement. What does spatially and temporally explicit mean in this context? Do you mean a high density of points in time and space or precise information about when and where samples were collected?

143 'downstream impacts extending into the Rio Grande'

152 Spatially explicit? Again, does this just mean at fixed locations? Seems fairly obvious.

153-156 Please provide context of responses to pre-fire storms and other seasonal DO fluctuations originating from Peralta Canyon rather than the Cochita reservoir outlet. Also, please include information about what this range of DO change and duration of effects means for aquatic biota. How long are were periods of 0 mg/L, and how rare are these prior to the fire?

156 What size events triggered these events and how were these distributed seasonally? Just following summer monsoon storms? How many of these neat lag events were tracked down stream? Was the model based on multiple events? It looks like 2 to 4 dates per location (Fig S2).

159 It seems that relations would vary by analyte with

164 State what the distance is to the confluence with Rio Puerco

181 Justify how estimates that do not account for regional differences are useful.

190-193 This is a powerful comparison and this context regarding reporting of stream length should be developed earlier.

212 Lack of 'spatially explicit post-fire water quality analysis?' Do you mean that previous researchers did not know where they were sampling? Don't you actually mean spatially structured, nested within stream networks or some approach that allows evaluation of downstream extent of wildfire effects?

S1 State the minimum fire size included in the annual tally.

Line 266 What information is there about how severity varied among fires, ecoregion and overtime

December 3, 2020

Dear Editor:

Please find enclosed the response to the reviewers' comments on our manuscript "Wildfires increasingly impact western US fluvial networks". In this letter, you will find the reviewers' comments and questions in *italics* and our responses in non-italicized fonts.

Reviewer #1 (Remarks to the Author):

The authors present a study that investigates the impacts of wildfire on fluvial networks in the western US by taking into account stream length impacted by burn events. I have not yet seen any published works that address stream water quality in this way, and thus the proposed work brings a fresh, novel perspective to wildfire science fields. The authors also use a Rio Grande case study to assess downstream propagation effects of wildfire and scale these data to the greater western US region. The authors provide a convincing argument for propagation of wildfire ecological effects in river networks; however, the following comments should be addressed before I can recommend this manuscript for publications in Nature Communications:

- Figure 1 – While I appreciate the graphical visualization of regional wildfire distribution in this way, I had a hard time discerning different ecoregions using the current color scheme. Some colors are very similar to one another (e.g., WCSP and WSMP). Please consider changing some of these colors so ecoregions can be easily identified.*

Response: We appreciate this comment, and agree that these two ecoregions are hard to tell apart. We have changed the color for WSMP to a distinct shade of green to help differentiate it from the other ecoregions. This color scheme adjustment has also been applied to Figure 4.

- L114-120 – The authors refer to Table S3 when describing ecoregion-specific regressions of SL_{ba} versus area burned. Some of these relationships were highly significant and I was curious to see how the data were distributed – p values and R^2 values only tell part of the statistical story. I recommend that you provide plots for each ecoregion in the supporting information so readers can quickly visualize these relationships.*

Response: We acknowledge the importance of showing these data graphically, and have generated Figure S3 to show ecoregion-specific regressions of SL_{ba} vs area burned, as well as accompanying density plots showing the ecoregion-specific distributions of both variables along their respective axes.

- L146-148 and Figure 3 – Did these DO sags coincide with storm events or were they purely a result of the burn itself? I'm curious to know whether hydrology played a role in driving DO dynamics. If the DO sags coincide with storm events, how does the magnitude of DO depletion compare with non-fire periods?*

Response: L171-184 of the resubmitted manuscript now include a more detailed description of DO conditions before and after the fire to provide context for how unusual were the DO sags reported. These statistics include the magnitude, frequency, and duration of DO sags pre-fire and post-fire, which are further broken down into monsoon and non-monsoon categories. We also include

additional information about the notable lack of DO sags directly below Cochiti Dam during the post-fire monsoon period.

- L152-153 – *A more detailed description of the DO exponential decay model is needed. Why did the authors select this particular statistical approach for modeling longitudinal propagation?*

Response: We used an exponential model because it offers a strong statistical prediction of our dataset, and also provides a mechanistic interpretation of transport processes in rivers. Specifically, an advection-reaction transport model describing the spatial variation of a reactive substance can be cast as:

$$\frac{\partial C}{\partial t} = -U \frac{\partial C}{\partial x} - k_r C$$

where C is the concentration of the reactive substance, t is time, U is the mean flow velocity (note that $x = U t$) and k_r is the reaction rate coefficient. At steady-state, $C = C_0 \exp(-\frac{k_r}{U} x)$, which represents spatial exponential decay. L183-187 of the resubmitted manuscript now state: “We analyzed the longitudinal propagation of distinguishable DO sags generated after four monsoon runoff events that occurred after the Las Conchas wildfire and fitted an exponential model to estimate the extent of these wildfire impacts on water quality. We chose this model because it provided a strong statistical prediction of our dataset, and a mechanistic interpretation of advection and reaction processes in rivers⁴⁷.”

- L156 – *“persisted 344 km downstream” ... What is the error on this number? Please also include error bars or shading in Figure S2. How realistic is the extrapolation of longitudinal distance from the data shown in Figure S2?*

Response: We appreciate this comment and agree that it is important to represent the uncertainty associated with the fit line and our interpretation of it. To that end, we constructed 95% confidence intervals for the exponential decay fit line to assess the error associated with the estimate of 344 km. Using the intervals, which are now included as thinner red lines in Figure S2, there is a 95% chance that the fit line signifying downstream decreases in DO reach the 0.5 mg/L threshold between 235 and 552 km downstream. While there is uncertainty associated with our estimate of SL_{LE} , we note that the confidence interval suggests that the value is likely higher than the fit estimate of 344 km. We have incorporated the uncertainty associated with this fit into Figure 4 as error bars.

- L162 – *I know it is described in the methods, but Horton’s law of stream order needs a brief description and reference here too.*

Response: Thanks for noting this important issue. Given the journal’s presentation style, we initially decided to keep the bulk of the description in the Methods section but now have explained the essence of the work done in the main body. L193-199 of the resubmitted manuscript now read “As an independent verification of this estimation, we repurposed Horton’s law of stream length⁴⁸ to create a mathematical framework commensurate with our data availability to validate our estimates of SL_{LE} . This work resulted in a new longitudinal model to estimate SL_{LE} downstream of burned areas. When applied to our dataset, it estimated that the disturbances could propagate to an 8th order stream (i.e., the Rio Grande downstream of its confluence with the Rio Puerco, 197 km downstream of the headwaters of Peralta Canyon, Figure 3A; see Methods).”

- L216 – *“Wildfires are a top-10 contributor” ... According to whom? Needs a reference. Could also say a “significant contributor”.*

Response: We added the following key references to substantiate this claim:

39. USEPA. National Summary of Impaired Waters and TMDL Information. https://iaspub.epa.gov/waters10/attains_nation_cy.control?p_report_type=T.
 49. Murphy, S. F., McCleskey, R. B., Martin, D. A., Writer, J. H. & Ebel, B. A. Fire, Flood, and Drought: Extreme Climate Events Alter Flow Paths and Stream Chemistry. *J. Geophys. Res. Biogeosciences* 123, 2513–2526 (2018).
 50. Murphy, S. F., Writer, J. H., McCleskey, R. B. & Martin, D. A. The role of precipitation type, intensity, and spatial distribution in source water quality after wildfire. *Environ. Res. Lett.* 10, 084007 (2015).
 51. Neary, D. G., Klopatek, C. C., DeBano, L. F. & Ffolliott, P. F. Fire effects on belowground sustainability: a review and synthesis. *For. Ecol. Manag.* 122, 51–71 (1999).
- *L229-230 – Stationary monitoring are still essential for water quality assessments. Perhaps rephrase to suggest that dynamic monitoring stations are also needed and can leverage data collected continuously from existing fixed-location monitoring stations.*

Response: We completely agree with the reviewers’ point that both approaches are needed. L269-273 of the resubmitted manuscript now read: “Thus, we must begin to incorporate ‘dynamic’ monitoring approaches focused on longitudinal data collection that supplement traditional ‘stationary’ ecological monitoring strategies (e.g., CZO and LTER) to fully characterize the extent of wildfire-driven water quality excursions along fluvial networks. Dynamic monitoring should be the priority of rapid-response teams.”

- *Finally, the impact of predicting longitudinal effects of wildfire is not limited to drinking water, but these findings also have major implications for rivers draining coastal mountainous regions (key geographic feature of California) – The data presented here suggest that downstream propagation of wildfire could alter the amount or composition of carbon, nutrients, or other materials entering estuarine and coastal ocean environments. It would be great if the authors could expand on this in the conclusion.*

Response: Thanks for this suggestion. L286-289 of the resubmitted manuscript now reads: “These disturbances mobilize sediments, nutrients, carbon, metals, and other environmentally relevant materials, affecting water quality dynamics not only along fluvial networks but also in downstream lacustrine, estuarine and marine environments.”

Reviewer #2 (Remarks to the Author):

General questions

- *What are the noteworthy results? Yes, the manuscript certainly provides noteworthy results on the extent of the western US river networks impacted by wildfire.*
- *Will the work be of significance to the field and related fields? How does it compare to the established literature? If the work is not original, please provide relevant references. The work is original, and there’s not much on this topic. There are hundreds or thousands of manuscripts on the fire impact on communities, sediment dynamics, and biogeochemistry, but there are none assessing the extent. I believe it might impact the field, although a miss a proper discussion of the drivers, consequences and meaning.*

- Does the work support the conclusions and claims, or is additional evidence needed? The work support the claims, there's no need for additional evidence.

- Are there any flaws in the data analysis, interpretation and conclusions? - Do these prohibit publication or require revision? No, I believe this is a very good numerical exercise, with no apparent flaws.

- Is the methodology sound? Does the work meet the expected standards in your field? Yes, I believe that the methodology is sound.

- Is there enough detail provided in the methods for the work to be reproduced? Yes, I believe that the methods are properly described and the work is therefore reproducible.

- I have a general comment, and some more specific comments. The general is rather conceptual, as I wonder if wildfires are a threat? I mean, they would be only a threat if wildfires were occurring at a frequency, severity and extent higher than what would occur in natural conditions. Fires, as floods, or extreme droughts, are extreme events that have occurred for millennia in western fluvial networks, and the local biodiversity is adapted to it. You need to further discuss this aspect, as perhaps we're recovering a natural disturbance that will allow the communities to recover their pre-settlement composition. In this direction, I would see the fires in the Amazon as a threat, but perhaps not the fires in central Australia or western US. Perhaps wildfires are a short-term threat for water supply, as drinking water treatment facilities relying on surface water might be imperiled, but this is an impact on humans, not on the ecosystem itself.

Response: Thank you for this thought-provoking comment and question. We agree, wildfires can be naturally occurring, and some aquatic communities have evolved with this disturbance. However, we believe that wildfires do constitute a threat to aquatic ecosystems for two reasons. First, due to historical land management practices and climate change, wildfires are increasing in frequency, severity, and extent across the globe in multiple biomes, as stated in the first and last sentence of the first paragraph (L30-36 of our resubmitted manuscript). Thus, while wildfires are natural phenomena, they are currently occurring at rates and intensities beyond which local biotic communities have evolved, causing previously unexperienced disruptions to hydrologic, nutrient, and carbon cycles in terrestrial and aquatic ecosystems. Thus, quantifying the stream length impacted by fires, and how this has changed over the past 30 years is highly relevant. Second, while some biological communities may have adapted to wildfire disturbances, a larger share of the human population is becoming increasingly vulnerable to wildfire impacts, as stated in the second paragraph of this manuscript (L37-47 of our resubmitted manuscript), and as we noted with the discussion of the long closure of Albuquerque's water treatment plant after the Las Conchas wildfire.

Through our manuscript, we have used the word 'threat' with a careful explanation of what we mean and thus believe that no changes are needed; for example:

⋮ “The 2019-20 fire season in Australia was the worst in recent history³, wildfire activity in the Amazon is **threatening** to shift the region from a sink to a source of carbon,…”

⋮ “Thus, fire damage to aquatic systems in forested watersheds represents a significant and costly **threat** to water security, both regionally for the western US and globally¹⁸⁻²⁰. “

⋮ “Despite the growing **threat** of wildfire impacts to fundamental aquatic ecosystem services that protect public health, including water purification¹⁷, flood mitigation³¹, and retention of sediments and nutrients^{32,33}, we lack quantitative estimates of the stream length directly impacted by wildfire (i.e., within burned areas) and how far downstream fire-driven water quality impacts propagate.”

⋮ “Wildfires are a top-10 contributor to water quality impairment and represent a significant **threat** to water security.”

Specific comments

- *P2, L40. Nature is an international journal, please use internationally accepted units for surface.*

Response: Thanks for catching this. We have now used square kilometers.

- *P2, L43. Please, develop more the point on water security. Is it for the changes in the suspended solids (ashes) during the first few months after the fire? Or for what? In terms of water supply, the amount of flowing water might be higher owing to reduced ET in the burned areas. Please, discuss this point. What stated in lines 45-49 is not enough to justify your point.*

Response: Thanks for suggesting these changes, which we agree will make this section stronger. The new text in L48-63 of our resubmitted manuscript now reads “There is growing evidence that wildfires trigger cascading impacts in fluvial networks over a range of spatiotemporal scales^{18,21–24}. Wildfires originate on hillslopes and cause decreased infiltration capacities and groundwater recharge^{25–27}, increased overland flow^{22,28}, reduced flood attenuation capacity by riparian vegetation²⁹, increased snow ablation³⁰, and higher frequency of landslides, avalanches, and debris flows^{31,32}. Post-fire precipitation events mobilize wildfire generated material from terrestrial ecosystems into streams and rivers within burned areas, which in turn drain into larger fluvial networks. Along these pathways, surface water quality drastically changes due to increased fluxes of ash, sediments, nutrients, carbon, and metals, commonly causing exceedances to limits set by the World Health Organization’s safe drinking water standards^{17,33–36}, and increasing costs associated with irrigation and drinking water supply^{37,38}. Wildfire disturbances also contribute to at least ten of the top twenty most critical disturbances listed in the US Environmental Protection Agency’s Clean Water Assessment (US EPA CWA³⁹), i.e.: elevated sediments, nutrient enrichment, organic enrichment and oxygen depletion, elevated temperature, elevated metal concentrations, habitat alterations, elevated turbidity, flow alterations, elevated salinity and/or total dissolved solids, and changes to pH and conductivity^{17,40,41}.”

- *P4, Figure 2. Is the temporal trend in the SL_{BA} related with increase in the forest cover? With a decrease in rainfall? Regardless of the driver, I suggest you use this figure to discuss the general comment on what is a disturbance and what not? I mean, perhaps we’re currently surpassing the natural disturbance fire regime.*

Response: The authors appreciate this comment, and agree that changes in forest cover and rainfall could potentially be important drivers of temporal SL_{BA} trends we report. Although deciphering the likely suite of drivers of SL_{BA} is beyond the scope of our study, we provide a preliminary discussion of these drivers below. For forest cover, we refer to the United States Forest Service report FS-1035, which describes changes in forest cover over the study period for the Western US. First, we note a geographical discrepancy between our definition of the Western US, which excludes Alaska, Hawaii, North and South Dakota, Nebraska, and Kansas, and FS-1035, which includes them. Between 1987 and 2012 (the year reported in the table on FS-1035 Pg. 8 that is closest to the first and last years of our data, respectively), forest land area increased from 333 to 346 million acres, i.e., the 2012 value represents an increase of ~4% over the 1987 value. If we take SL_{BA} values from 1987 and 2012 from Table S2, we get an increase of 333%. However, we note that 2012 was a relative outlier representing

the largest annual SL_{BA} value in our dataset. If instead of the 2012 SL_{BA} value, we apply the slope we calculated in Figure 2 (an increase of 342 km / year) to the 1987 SL_{BA} value, we estimate a SL_{BA} of 16,435 km for 2012, which is equivalent to an increase of 108%. Thus, while we note that forest cover increased in the Western US, it is unlikely that this mechanism can explain the observed increase in SL_{BA} , which is more than an order of magnitude greater. An equivalent analysis of rainfall is more difficult, because the relationship between rainfall and wildfire likely varies based on amount, intensity, and timing. Instead, we reference Holden et al. (2018) (<https://doi.org/10.1073/pnas.1802316115>), who found that rainfall is likely a primary driver of increased wildfire burn area, which we find correlates strongly to SL_{BA} in most ecoregions (Table S3). Because changes in rainfall are likely linked to changes in global climate, we find it probable that the increase in SL_{BA} is indeed associated to climate change, and we are likely surpassing the natural disturbance fire regime, as suggested by Holden et al. (2018) as well as studies referenced in the manuscript (e.g., Abatzoglou & Williams, 2016; Westerling et al., 2006; Westerling et al., 2011).

Reviewer #3 (Remarks to the Author):

General Comments

- *The water quality responses to wildfires and the associated changes in ecosystem services from burned watersheds are generally well appreciated. It will be less clear to many readers, why stream length is a relevant metric for gaging them. Stream length is a common reporting structure for water quality impairment but the link to wildfires requires some justification and explanation. Wildfire impacts originate from burning of vegetation and organic soil layers, typically in upland areas and sometimes in riparian vegetation adjacent to streams. Their effects do not originate from changes in streams themselves and evaluating the relative magnitude of effect is typically based on the area burned. Watershed effect of wildfires are usually assessed at the outlet of a burned watersheds, so there is need to describe how stream length and catchment areas are related across ecoregions and how those patterns may vary after fire. To help justify how SL_{BA} may have values as a metric of wildfire effects, it would be helpful to include a quick summary of the common sources of stream degradation (nutrients, sediment, temperature, etc.) and how they vary with relevant ecoregional differences. Currently, the authors arrive at this sort of comparison near the end of the paper. This general context should be presented at the very beginning of the paper, so readers will grasp the regulatory relevance for reporting stream length.*

Response: Thanks for highlighting the need to be more explicit about how wildfire impacts to stream water quality are related to SL_{BA} and SL_{LE} and how these metrics are useful. We added the following text in paragraphs three and four of the introduction to do so:

“There is growing evidence that wildfires trigger cascading impacts in fluvial networks over a range of spatiotemporal scales^{18,21–24}. Wildfires originate on hillslopes and cause decreased infiltration capacities and groundwater recharge^{25–27}, increased overland flow^{22,28}, reduced flood attenuation capacity by riparian vegetation²⁹, increased snow ablation³⁰, and higher frequency of landslides, avalanches, and debris flows^{31,32}. Post-fire precipitation events mobilize wildfire generated material from terrestrial ecosystems into streams and rivers within burned areas, which in turn drain into larger fluvial networks. Along these pathways, surface water quality drastically changes due to increased fluxes of ash, sediments, nutrients, carbon, and metals, commonly causing exceedances to limits set by the World Health Organization’s safe drinking water standards^{17,33–36}, and increasing costs associated with irrigation and drinking water supply^{37,38}. Wildfire disturbances also contribute to at

least ten of the top twenty most critical disturbances listed in the US Environmental Protection Agency's Clean Water Assessment (US EPA CWA³⁹), i.e.: elevated sediments, nutrient enrichment, organic enrichment and oxygen depletion, elevated temperature, elevated metal concentrations, habitat alterations, elevated turbidity, flow alterations, elevated salinity and/or total dissolved solids, and changes to pH and conductivity^{17,40,41}.

Despite the growing threat of wildfire impacts to water quality and ecosystem services that protect public health, we lack quantitative estimates of the stream length directly impacted by wildfires (i.e., within burned areas) and how far downstream wildfire-driven water quality impacts propagate. Addressing these knowledge gaps is important to 1) alert downstream communities about the risk of exposure to wildfire-related contaminants, 2) anticipate the range of potential impacts of wildfire to downstream water quality, 3) develop risk maps to mitigate the damage of property and infrastructure in fire-prone areas, 4) implement water treatment techniques capable of removing wildfire-specific physical and chemical pollutants, 5) build ecological infrastructure (e.g., fish by-passes, debris retention structures) to promote ecosystem resilience and recovery, and 6) design effective restoration projects to increase water filtration and sediment retention at local to regional scales.”

- *The effects of wildfire on downstream water quality relate to the absolute and proportional extent and severity of a given wildfire within a catchment. Wildfires create a complex mosaic of effects since they burn forests and effect watersheds to varying extents depending on the specific patterns and proportional extent of high, moderate, low severity, as well as unburned areas. This is especially relevant to the riparian and stream corridor, since these parts of the landscape may burn at different frequency or severity than upland areas. The authors state that they consider burn severity, but that it was unclear how that information was used. As background, it would be useful to know how the relation between burn severity and fire size varies within ecoregion has that changed with time? What is the relation between proportional burn severity and wildfire extent? Additionally, some characterization of the pre-fire variation in stream density by ecoregions would be a useful precursor before evaluating how wildfire effects stream length regionally and how those changes have varied with time.*

Response: We agree with the reviewer in that explicitly including burn severity in analyses of wildfire propagation would be ideal to improve predictions pertaining to water quality impacts. In our work, we used fire severity to include only areas of wildfires with low, medium, or high burn severity (as defined in the Methods), but not to understand how fire severity affects water quality outside of burned watersheds, simply because we do not have half of the information needed to do so. Thus, while we (government agencies, scientists and practitioners) currently map what happens inside burned areas, we do not have active programs collecting the other half of the information that is needed to complete the puzzle, i.e., how what happens in burned areas (severity and extent) translates into different degrees of impacts on water quality dynamics in fluvial networks. We have dedicated the last part of the manuscript to raise awareness on the need to collect the second half of the information needed to gain a more meaningful understanding of wildfire impact propagation and their impact on water quality beyond burned areas.

- *The downstream extent of post-fire water quality responses is very interesting. It would be great to know how this relates to the size (proportional extent by fire severity), the size of the watershed, stream discharge, time since fire, etc. The Las Conchas Fire study is interesting, but there is no justification for extrapolating this limited local information across the western US. It would be useful to learn more about whether the observed post-fire DO responses vary with*

storm characteristics (precip amount and intensity), stream discharge and temperature and also how they differ from pre-fire conditions can DO responses to storm events.

Response: We agree that while our study case may be thought of as data-limited and localized, it represents, to date, the only dataset that has captured the propagation of a major wildfire (i.e., second largest recorded in New Mexico, US) along spatial scales relevant to understanding the longitudinal propagation of wildfire disturbances (i.e., hundreds of kilometers downstream of the burned area), over multiple years before, during and after the wildfire. Because of this comment, and similar comments from other reviewers, we now include more explicit information about the hydrology associated with DO sag events, as well as additional information about pre-fire DO behavior in L171-179 for context. Also relevant, the disturbances caused by the Las Conchas wildfire directly affected for three months the supply of drinking and irrigation water to ~1 million people living in the Albuquerque metropolitan area, which adds a special angle to improving the understanding of societally relevant impacts of disturbances to hydrologic systems. In L245-280 of the resubmission file, we make these points clear and emphasize the importance of starting to collect longitudinal data to improve our understanding of how we can relate characteristics of the burned areas, the hydrology of the watershed and expected water quality outcomes. In this context, we believe that our manuscript contributes to exploring these critical points and can help engage scientist and practitioners from government, university and consulting areas to recognize the need to collect better data on watershed disturbances and their effects on ecosystem services.

Specific Comments

- *Line 47 This is incorrect and confusing. Wildfire impacts originate from the burning of vegetation, typically in upland areas surrounding streams and does not originate from changes in streams themselves. This is likely referring to ‘the cascading impacts in fluvial networks’ caused by wildfire. Nonetheless, wildfires trigger these by altering terrestrial ecosystems and the emergent responses scale with the area of terrestrial ecosystems burned at known levels of wildfire severity. The relations between burn area and stream length may be interesting to consider and report and how se can be proportional area burned After the initial erosion-related responses, post-fire stream nutrient export is the response of lack of plant uptake and accelerated nutrient supply from exposed burned areas. Reporting wildfire effect on a per length basis confuses this issue.*

Response: Thanks for catching this confusing description. To address your concerns and those raised by reviewer #2, we have modified this paragraph in lines 48-63 of the resubmitted manuscript.

- *58 A portion of the Rio Grande in New Mexico? Where? How long? Why? The river does not originate or end in New Mexico, so a bit of clarity would be helpful for many readers?*

Response: Thanks for this suggestion. Given that this text is in the Introduction and we later discussed the study case in detail, we updated the text in L 78-82 of the resubmitted manuscript to read: “Next, we use high-frequency in-situ sensor data capturing the spatial propagation of post-fire water quality disturbances in the Rio Grande (New Mexico, US) to estimate the Stream Length: Longitudinal Extent (SL_{LE}) impacted by an individual wildfire as a case study, and upscale this approximation to the study region.”

- *69 Specify the size catchments and minimum stream order included in the survey.*

Response: For this analysis, we included all catchments and stream segments present in the NHD 1:24,000 layer, and did not exclude any catchments or streams based on size or order. We have added this information to the Methods (305-313 of the resubmission) to clarify.

- *70 Drier areas with few streams and more fires vs wet area with more streams and fewer fires.*

Response: We agree that the percentages presented in original L70 are likely a function of both precipitation and stream density. We have presented the ratio of total stream length (km) to total area (km²) within each ecoregion as a first approximation of stream network density across ecoregions in Figure S1. We have also updated this figure to include average annual precipitation (as color-coding).

- *73 'hotspot' meaning?*

Response: We decided to simplify the sentence to avoid ambiguity. The sentence now reads: “In this area, composed primarily of Cold Desert and Western Cordillera ecoregions, streams are disproportionately impacted by wildfire.” (see L94-95 in the resubmitted manuscript).

- *79 Can you report these with regard to stream density and area burned*

Response: We have summarized the relationships between stream network density and burned stream density in Figure S1, and summary statistics for area burned in Table S1.

- *90 Information on what fires are/aren't mapped is not clear obvious in Suppl Matter and site selection needs additional clarity.*

Response: We used publicly available fire extent and severity layers to guide our fire selection. All fires were included unless they were either located outside the geographic scope defined in Figure 1, or burned an area less than ~4.1 km². We have added this information to the Methods (305-313 of the resubmitted manuscript).

- *116-117 This relationship is getting at something useful. Is the difference across regions due to fire, vegetation and fuel relations or merely due to the differences in stream length network density?*

Response: While analyzing vegetation type and fuel resources are outside the scope of our study, it is likely that climatological differences that define unique ecoregions (like precipitation, as discussed in our response to Reviewer 2) likely alter both vegetation type and resulting fuel resources.

- *126 This would be expected, so how is this insightful?*

Response: Thanks for the comment. We believe that we offered an explanation for this after the sentence that you are referring to in L146-152 of the resubmission: “This suggests that the density of streams burned **generally** correlated with the density of streams in an ecoregion. **However**, ecoregion-specific factors (e.g., the Marine West Coast Forest ecoregion receives 210% of the annual precipitation of the next wettest ecoregion included in the study) created an outlier. Thus, the relationship between area burned and SL_{BA} is likely a function of regional factors, including

geomorphology, vegetation, and climate, suggesting that numerous factors control the extent to which wildfires impact aquatic resources in a given location.”

- 132 Explain what you mean by resilience and what this statement about ecoregional differences is based upon.

Response: We decided to drop the use of resilience as the key points that we intended to make were made clear before. Your point about ecoregional differences made us realize that we had not explicitly defined ecoregions and thus we added the following contextual sentence near the first use of the word ecoregion (in the Introduction; L76-78 in the resubmission): “We explore temporal SL_{BA} trends and relationships to the burned area as a function of established ecoregions⁴², since many drivers of wildfire, such as climate regime⁴³, drought⁴⁴, and snowpack⁴⁵, are ecoregion-specific.”

- 141 Provide a few salient points to support this statement. What does spatially and temporally explicit mean in this context? Do you mean a high density of points in time and space or precise information about when and where samples were collected?

Response: Thanks for this suggestion. The new text in L160-163 now reads: “The dataset, originally presented in Dahm et al.²² and used here, represents one of the most spatially and temporally explicit records of pre- and post-fire water quality available to date, as it registered the variation of water quality parameters along ~50 km of the Rio Grande, and over multiple years before and after the wildfire.”

- 143 ‘downstream impacts extending into the Rio Grande’

Response: We updated the text to read (L163-165 of resubmission): “The Las Conchas fire burned in the Western Cordillera ecoregion and propagated through portions of the Rio Grande in both Western Cordillera and Cold Desert ecoregions.”

- 152 Spatially explicit? Again, does this just mean at fixed locations? Seems fairly obvious.

Response: Since we clarified our meaning in previous lines, in response to a previous comment, we deleted “spatially explicit” here.

- 153-156 Please provide context of responses to pre-fire storms and other seasonal DO fluctuations originating from Peralta Canyon rather than the Cochita reservoir outlet. Also, please include information about what this range of DO change and duration of effects means for aquatic biota. How long are were periods of 0 mg/L, and how rare are these prior to the fire?

Response: In L169-182 in the resubmitted manuscript, we provide a more detailed description of the pre-fire record during the monsoon season; DO conditions outside of the monsoon season, both pre- and post-fire; the magnitude, frequency, and duration of DO sags observed during the monsoon season of 2011; DO conditions during the 2011 monsoon season immediately downstream of Cochiti Reservoir; and the persistence of DO sags post-fire.

- *156 What size events triggered these events and how were these distributed seasonally? Just following summer monsoon storms? How many of these neat lag events were tracked down stream? Was the model based on multiple events? It looks like 2 to 4 dates per location (Fig S2).*

Response: We appreciate this comment, and now provide a more detailed description of the pre-fire record. In the resubmitted manuscript (L 169-187), we include the magnitude of these events ($\sim 10\text{-}20\text{ m}^3\text{s}^{-1}$) and seasonal distribution (i.e., only during the monsoon season). We also explained that the longitudinal propagation of four post-fire DO sags were included in the exponential decay model, with each storm event including data from at least three of the four sites.

- *159 It seems that relations would vary by analyte [with]*

Response: We agree with this comment, and now provide contextual text to explain this point. These relationships are likely analyte-specific and that is why we have provided contextual text to explain this point (L190-192 in resubmission): “Since DO is reincorporated via reaeration during transport, we predict that the longitudinal propagation of more conservative signals (i.e., non-limiting nutrients, metals, or ash) extend farther than DO sags.”

- *164 State what the distance is to the confluence with Rio Puerco*

Response: The distance from the start of the Peralta watershed to the confluence of the Rio Grande and Rio Puerco is 197 km, which we have now included in the manuscript (L199 and L361 in the resubmission file).

- *181 Justify how estimates that do not account for regional differences are useful.*

Response: Thanks for giving us this action item. We modified the text in L 229-237 to read: “We recognize that this first approximation does not account for variability in watershed and stream conditions that could influence the longitudinal propagation of wildfire impacts, or for inter-ecoregion variability (i.e., Figure S1). Nevertheless, our estimates are a preliminary assessment that can be temporarily used to address problems requiring an answer to the question: how far do wildfire disturbances propagate along fluvial networks? More importantly, we emphasize that the limitation of our estimates serves as a wake-up call for the scientific community to start gathering longitudinal data to improve our predictive ability to assess disturbance propagation (see below our discussion about *Incorporating streams and rivers into fire science*).”

- *190-193 This is a powerful comparison and this context regarding reporting of stream length should be developed earlier.*

Response: Thanks for this comment. Based on your review and a similar comment from Reviewer #2, we have added a new paragraph in the introduction (L48-63 in the resubmission file) to make these points much clearer throughout the manuscript: “There is growing evidence that wildfires trigger cascading impacts in fluvial networks over a range of spatiotemporal scales^{18,21-24}. Wildfires originate on hillslopes and cause decreased infiltration capacity and groundwater recharge²⁵⁻²⁷, increased overland flow^{22,28}, reduced flood attenuation capacity by riparian vegetation²⁹, increased snow ablation³⁰, and higher frequency of landslides, avalanches, and debris flows^{31,32}. Post-fire precipitation events mobilize wildfire-generated material from terrestrial ecosystems into streams and rivers within burned areas, which in turn drain into larger fluvial networks. Along these pathways, surface water

quality drastically changes due to increased fluxes of ash, sediments, nutrients, carbon, and metals, commonly causing exceedances to limits set by the World Health Organization's safe drinking water standards^{17,33-36}, and increasing costs associated with irrigation and drinking water supply^{37,38}. Wildfire disturbances also contribute to at least ten of the top twenty most critical disturbances listed in the US Environmental Protection Agency's Clean Water Assessment (US EPA CWA³⁹), i.e.: elevated sediments, nutrient enrichment, organic enrichment and oxygen depletion, elevated temperature, elevated metal concentrations, habitat alterations, elevated turbidity, flow alterations, elevated salinity and/or total dissolved solids, and changes to pH and conductivity^{17,40,41}.

- *212 Lack of 'spatially explicit post-fire water quality analysis?' Do you mean that previous researchers did not know where they were sampling? Don't you actually mean spatially structured, nested within stream networks or some approach that allows evaluation of downstream extent of wildfire effects?*

Response: Thanks for this important note. In response, we have changed the text (L251-253 in the resubmission file) to read: "We attribute the lack of previous estimates to the notable absence of water quality data collected from multiple sensors located over spatial scales commensurate with the propagation of wildfire disturbances (i.e., over hundreds of kilometers)."

- *SI State the minimum fire size included in the annual tally.*

Response: We included all fires with a total burn size larger than 1,000 acres (~4.1 km²); this description is now included in the Methods.

- *Line 266 What information is there about how severity varied among fires, ecoregion and overtime*

Response: The fire severity layer we used to filter unburned areas out of our calculations provides comprehensive information about fire severity. While a comprehensive analysis of fire severity is outside the scope of this manuscript, we note that there are several reports that fires in this region are increasing not only in size, but potentially also in severity (e.g., Miller and Safford (10.4996/fireecology.0803041); Picotte et al. 2016 (<https://doi.org/10.1071/WF15039>)). How the impact of stream network density may impact fire intensity, for instance, could be a very interesting avenue for future research building off of the SL_{BA} framework we present.

Sincerely,

Dave Van Horn and co-authors

REVIEWERS' COMMENTS

Reviewer #1 (Remarks to the Author):

The authors have sufficiently addressed my initial feedback and I do not have any additional comments. Therefore, I recommend this manuscript for publication.

Reviewer #2 (Remarks to the Author):

Dear authors and editor

I have read the new version of the document, as well as the rebuttal, and I believe that the manuscript is now ready for acceptance. The only discrepancies I have are rather a matter of style and opinion, but I believe them not to be significant.

Best regards

Reviewer #3 (Remarks to the Author):

Comments for the Authors

The authors have done a fine job in addressing the previous review remarks. The paper is well-written and contains novel and interesting results. The following comments are minor and editorial in nature, as the article is now close to acceptable for publication.

Abstract

Specify what attributes of streams impact and impairment were consider

Specify the size (order?) of streams included in this analysis

Line 42 The 2020 fires are no longer on-going. These data can be updated with final numbers.

Line 71 I would recommend linking these stream responses to the immediate post-fire emergency prioritization of watershed responses and mitigation treatments that are conducted after wildfires on federal (BAER) and private lands (EWP) land. These aim to reduce sediment, nutrients, C and metals from hillslopes to streams, and consider longer term soil productivity and riparian and upland revegetation.

Line 87 Since this is under consideration in a top-tier journal, it probably goes without saying that the analysis is unique. Repetition on line 249 is also unneeded.

Figure 1 The figure is better but remains difficult to interpret. Are both the location and colors on the Density traces above and right of the map supposed to correspond to specific ecoregions? This seems likely to be the case but does not adequately clear in some instances. For example, the light green peak at the bottom of the right-hand trace (Gila Mtns?) is much lighter than what is probably the corresponding patch on the map in SE AZ and the boot of NM. Perhaps make the map colors 'transparent' to better match those on the Density plots.

Please include a date for the fires included on the map. Since the 2020 fires are now fully contained, were record setting, and are mentioned in the paper including these would make sense.

Tables

Are descriptive table captions presented elsewhere? Should they be?

Reviewer #1 (Remarks to the Author):

The authors have sufficiently addressed my initial feedback and I do not have any additional comments. Therefore, I recommend this manuscript for publication.

Response: Thank you so much for your insightful review. We truly appreciate your work.

Reviewer #2 (Remarks to the Author):

Dear authors and editor

I have read the new version of the document, as well as the rebuttal, and I believe that the manuscript is now ready for acceptance. The only discrepancies I have are rather a matter of style and opinion, but I believe them not to be significant.

Best regards

Response: Thank you so much for your insightful review. We truly appreciate your work.

Reviewer #3 (Remarks to the Author):

The authors have done a fine job in addressing the previous review remarks. The paper is well-written and contains novel and interesting results. The following comments are minor and editorial in nature, as the article is now close to acceptable for publication.

Abstract

- *Specify what attributes of streams impact and impairment were considered*

Response: For this study, streams impacted by wildfires were located within burned regions, as defined in the “Geospatial analysis” methods section. For longitudinal propagation of impacts downstream, we used the relationship developed in Figure S2 and a threshold of 0.5 mg/L to determine the longitudinal extent of impact, as we explained in the “Validating longitudinal stream length (SLLE) assumptions” Methods section. Because we believe that the sections referenced here adequately explain the impacts investigated, and because we are limited by the word count of the abstract, we have not changed our wording.

- *Specify the size (order?) of streams included in this analysis*

Response: We appreciate this request for clarification, and have replaced “stream length” with “stream+river length” across the manuscript to better communicate the range of systems considered in our analysis. We clarify that we used all stream orders present in the NHD dataset used in the “Datasets” methods section.

- *Line 42 The 2020 fires are no longer on-going. These data can be updated with final numbers.*

Response: Thanks for this suggestion. We updated the text to read: “It was recently superseded by the 2020 wildfire season¹⁵, which burned more than 17,000 km².”

- *Line 71 I would recommend linking these stream responses to the immediate post-fire emergency prioritization of watershed responses and mitigation treatments that are conducted after wildfires on federal (BAER) and private lands (EWP) land. These aim to reduce sediment, nutrients, C and metals from hillslopes to streams, and consider longer term soil productivity and riparian and upland revegetation.*

Response: Thanks for this suggestion. We added the following sentence: “5) implement postfire emergency watershed rehabilitation techniques to reduce the movement of sediment, burned vegetation, nutrients, metals, and other contaminants from hillslopes to streams, and 6) design

effective long-term postfire restoration projects to increase revegetation, water filtration, and sediment retention at watershed scales.”

- *Line 87 Since this is under consideration in a top-tier journal, it probably goes without saying that the analysis is unique. Repetition on line 249 is also unneeded.*

Response: Thanks for this suggestion. However, we want to keep the text as is in these lines because we want to highlight our concern about the status quo and convey a sense of urgency about the need for more studies on this topic.

- *Figure 1 The figure is better but remains difficult to interpret. Are both the location and colors on the Density traces above and right of the map supposed to correspond to specific ecoregions? This seems likely to be the case but does not adequately clear in some instances. For example, the light green peak at the bottom of the right-hand trace (Gila Mtns?) is much lighter than what is probably the corresponding patch on the map in SE AZ and the boot of NM. Perhaps make the map colors ‘transparent’ to better match those on the Density plots.*

Please include a date for the fires included on the map. Since the 2020 fires are now fully contained, were record setting, and are mentioned in the paper including these would make sense.

Response: We appreciate this feedback and have edited the transparency of both density plots to match the map and legend. We have also added dates to the Figure 1 caption. We have elected not to include the 2020 fire season because the data products used are continually updated, and would require a full redo of all analyses, statistics, tables, and figures, which is not feasible at this stage.

Tables

- *Are descriptive table captions presented elsewhere? Should they be?*

Response: We added descriptive table captions next to each table in the Supplementary File.

Sincerely,
Dave Van Horn and co-authors